# Novel Magnetite Nanocomposites (Fe₃O₄/C) for Efficient Immobilization of Ciprofloxacin from Aqueous Solutions through Adsorption Pretreatment and Membrane Processes

**Muhammad Zahoor** [1,*] **, Azmat Ullah** [2] **, Sultan Alam** [2] **, Mian Muhammad** [2] **, Roy Hendroko Setyobudi** [3] **, Ivar Zekker** [4,*] **and Amir Sohail** [5]

1  Department of Biochemistry, University of Malakand, Chakdara KPK 18800, Pakistan
2  Department of Chemistry, University of Malakand, Chakdara KPK 18800, Pakistan; azmat.ullah@sjtu.edu.cn (A.U.); dr.sultanalam@yahoo.com (S.A.); mianchem@uom.edu.pk (M.M.)
3  Department of Agriculture Science, University of Muhammadiyah Malang, Malang 65145, Indonesia; roy_hendroko@hotmail.com
4  Institute of Chemistry, University of Tartu, 14a Ravila St., 50411 Tartu, Estonia
5  MSC Construction Project Management University of Bolton, Bolton BL3 5AB, UK; syedaamirsohail89@gmail.com
*  Correspondence: mohammadzahoorus@yahoo.com (M.Z.); ivar.zekker@ut.ee (I.Z.)

**Abstract:** The release of antibiotics into the aquatic environment enhances the drug resistance capabilities of microorganisms, as in large water reservoirs, their concentrations are lesser than their minimum bactericidal concentration, and microorganisms living there become resistant to such antibiotics. Therefore, robust hybrid technologies, comprising of efficient conventional adsorption processes and modern membranes processes, are needed to effectively remove such pollutants from industrial effluents. The present study is an attempt where iron-based magnetic carbon nanocomposites (Fe₃O₄/C) were prepared from mango biomass precursors and utilized as an adsorbent for the removal ciprofloxacin from wastewater in combination with three types of membranes that are robust but fouled by organic matter. The Fe₃O₄/C composite was characterized using energy dispersive X-ray (EDX) technique, X-ray diffraction (XRD), scanning electron microscopy (SEM), Fourier transform infrared spectroscopy (FTIR), Brunauer Emmett Teller (BET), Barrett–Joyner-Halenda (BJH) surface area, Thermogravimetric (TG)/Thermal differential analysis (DTA) and point of zero charge pH analyses. Initially, batch adsorption experiments were used to determine adsorption parameters. Then the adsorption unit was coupled with membrane pilot plant where the adsorption role was to adsorb CIPRO before entering into the membrane unit to control fouling caused by selected antibiotic. In batch experiments, the equilibrium time was found as 60 min and kinetics data were more favorably accommodated with the pseudo-2nd-order model ($R^2 = 0.99$). Langmuir model ($R^2 = 0.997$) more favorably accommodated the equilibrium data in comparison to other models used such as the Freundlich ($R^2 = 0.86$), Temkin ($R^2 = 0.91$) and Jovanovich ($R^2 = 0.95$) models. The thermodynamic aspects of the adsorption process were also evaluated and the process was found to be spontaneous, feasible and exothermic. The influence of adsorbent dosage and pH, were also investigated, where the optimal adsorption conditions were: optimum pH = 7 and optimum Fe₃O₄/C dosage = 0.04 g. The CIPRO-loaded nanocomposite was regenerated with NaOH, CH₃OH and distilled water several times. Improved percent rejections of CIPRO and permeate fluxes with the membrane/adsorption operation were observed as compared to naked membrane operations. Magnetic adsorbent was found as a best solution of foul control; a defect in the modern robust technology of membranes. However, further experimentation is needed to validate the present findings.

**Keywords:** adsorption; membranes; fouling; permeate flux

## 1. Introduction

Pharmaceutical residues in water reservoirs are the emerging pollutants that have been reported by a number of researchers from various parts of the world. Among such residues, the antibiotics have a direct impact on human health because they are diluted in large bodies of water reservoirs and bacteria living there come in contact with its concentration that are lower than their minimum bactericidal concentration which thus develop resistance against such antibiotic. As a result, if a person is infected with that bacterium, they will not be cured by that antibiotic. This is a global issue and needs immense attention to devise methods for recovery of antibiotics from industrial effluents. Apart from industrial effluents there are other ways too that have share in polluting water reservoirs with antibiotics like inappropriate dumping of expired antibiotics, from the sewage and terrestrial landfilling of biosolids [1–3]. The overall responsible sources are enlisted in the following diagram (Scheme 1):

**Scheme 1.** Entry sources of antibiotics to the environment.

Among the antibiotics, the fluoroquinolones (FQs)-based antibiotics are the most persistent classes having a long half-life and low biodegradability [4]. Ciprofloxacin (CIPRO) belongs to this class of antibiotics that is a broad-spectrum antibiotic and used to treat both gram- positive and negative bacteria. It has been reported that its contribution in

drug resistance problem many folds as compared to other antibiotics due to it persistent nature in the environment [5]. Additionally, it has been detected in the aquatic environment from ng L$^{-1}$ to mg L$^{-1}$ range [6].

A number of methods have been applied for the removal of fluoroquinolones-based antibiotics from waste-water such as adsorption [7,8], advanced oxidation process [9], electrocoagulation [10] and membrane processes [8,11]. Among these methods, although adsorption is a preferably used method because of its low cost, high efficiency and lower environmental risk as associated with other methods [12], but being a conventional method, it cannot be used as robust approach in reclamation of portable water from industrial effluents. On the other hand, membranes being a robust technology that can be installed in line with the drainage/effluent line for cleaning water can be obtained with no waiting for long times but due to concentration polarization they are fouled by the contaminants which adversely affect the membrane efficacy and frequent back washes are required. Every membrane has its own molecular weight cut off values and molecules larger than that limit are accumulated near to membrane surface; a phenomenon known as concentration polarization. Some of these molecules get adsorbed on membrane surface thereby blocking the membrane pores; known as fouling [13–15]. As a result, low permeate flux is encountered. To solve this problem scientists have tried to couple membrane operations with adsorption pretreatment. Initially, activated carbon being an efficient adsorbent was applied in continuous stirred reactor installed ahead of membrane unit and it was considered to be a best solution being based on the assumption that the organic matter would get adsorbed onto activated carbon effectively. The organic pollutant loaded activated carbon upon gaining entry into the membrane unit would form a porous layer over the membrane surface and it was presumed that this layer would not affect the permeate flux. However, practically a decline in permeate flux was observed with activated carbon. As a solution to this problem, Zahoor et al. converted activated carbon into a magnetic activated carbon composite with an idea that due to the magnetic character the composite can be stopped from entering into membrane system simply by applying a magnet after loading with pollutant in the reactor. The traces of pollutant left in the solution (as adsorption processes are not 100% efficient) would not affect the membrane permeate flux too much as compared to naked membrane and activated carbon/membrane hybrid operations. Although a reduction in surface area occurred with impregnation of iron oxide over activated carbon but still its effectiveness in foul controlling was more as compared to its counterpart—activated carbon [8,16–19].

The use of magnetic adsorbent could be the ultimate solution to control fouling in membranes, however, there are very few reported studies on this topic. Furthermore, such studies are only limited to surfactants, nitrophenols, etc. To the best of our knowledge such studies has been not carried out on ciprofloxacin before. Therefore, the aim of this study was to synthesize iron-based carbon nanocomposites (Fe$_3$O$_4$/C) from mango biomass precursors and to estimate the ciprofloxacin removal efficiency from aqueous media in a reactor equipped with adsorption and membranes units. The prepared Fe$_3$O$_4$/C composite was characterized by various instrumental techniques. The magnetic adsorbent has been used with the aim that it would enhance membrane efficacies in directly removing pollutants from drainage lines without storing the effluents in collecting tanks as required in industries where adsorption units are installed.

## 2. Materials and Methods

### 2.1. Reagents and Materials

Biomass precursors of mango were collected from local market in Swat, KP Pakistan. Ciprofloxacin was collected from Swat Pharma (Figure 1). FeSO$_4$ × 7H$_2$O, FeCl$_3$ × 6H$_2$O, HCl, NaOH and humic acid were purchased from the Sigma-Aldrich (Taufkirchen, Germany). All the reagents and chemicals used in current research were with analytical grade (purity > 99%). Double-distilled water was utilized in the experiments.

**Figure 1.** General chemical outline structure of ciprofloxacin.

The characteristics properties of the membrane used are listed in Table 1.

**Table 1.** Characteristic properties of UF, NF and RO membranes.

| UF Membrane | | NF Membrane | | RO Membrane | |
|---|---|---|---|---|---|
| Parameters | Specification | Parameters | Specification | Parameters | Specification |
| Material | Polyether sulfone | Model | NF 270-2540 | Model | RO 270-2540 |
| Type | Capillary multi bores × 7 | Permeate Flow rate | 850 gallons/day (3.2 m$^3$/day) | Membrane type | Thin film composite (Filmtech) |
| Diameter bores ID | 0.9 mm | Active surface area | 28 ft$^2$ (3.2 m$^2$) | Permeate Flow rate | 850 gallons/day (3.2 m$^3$/day) |
| Diameter fiber OD | 4.2 mm | Applied pressure | 4.8 bar | Active surface area | 28 ft$^2$ (3.2 m$^2$) |
| Stabilized salt rejection | 10–20% | Stabilized salt rejection | >97% | Stabilized salt rejection | 100% |
| Surface area | 50 m$^2$ | Surface area | 3.2 m$^2$ | Surface area | 3.2 m$^2$ |
| Maximum temperature | 40 °C | Maximum temperature | 40–180 °C | Maximum temperature | 40–180 °C |
| Maximum pressure | 109 psi | Maximum pressure | 100–1000 psi | Maximum pressure | 100–1000 psi |
| Membrane back wash pressure | 0.5–1 bar | Membrane back wash pressure | 50–800 psi | Membrane back wash pressure | 50–800 psi |
| Operator pH range | 3–10 | Operator pH range | 3–10 | Operator pH range | 3–10 |
| Back wash pH range | 1–13 | Back wash pH range | 1–12 | Back wash pH range | 1–12 |
| Disinfection chemicals | Hypo chloride and Hydrogen peroxide | Disinfection chemicals | Hydrogen peroxide and peracetic acid | Disinfection chemicals | Hydrogen peroxide and peracetic acid |
| MWCO | 100 KD | MWCO | 0.3–1 KD | MWCO | 0.1–1 KD |
| Pore size | 5–20 nm | Pore size | 1–5 nm | Pore size | 1–5 nm |

### 2.2. Preparation of Magnetic Carbon Nanocomposites (Fe$_3$O$_4$/C)

Mango waste biomass based magnetic carbon nanocomposite was prepared using a co-precipitation method. A suspension of FeCl$_3$ × 6H$_2$O (0.05 mol) and FeSO$_4$ × 7H$_2$O (0.025 mol) was prepared by mixing 200 mL of each in water at room temperature. To which then mango waste biomass in crushed form was mixed where 100 mL of NaOH (5 molL$^{-1}$) was added dropwise at 70 °C within a time interval of 50 min. The final mixture

was then cooled. The mixture pH was brought to neutral through thorough washing with distilled water. The resulting neutral mixture was then ignited and charred in a specially constructed chamber in nitrogen atmosphere. The final product was oven-dried at 70 °C.

### 2.3. Characterization of Fe$_3$O$_4$/C Composite

The most substantial feature that directly contributes to the removal capacity of a pollutant by a given adsorbent is its porosity and surface area. Both these features are interlinked and are mandatory to be determined while dealing with adsorption studies. About 0.1 g sample of Fe$_3$O$_4$/C was subjected to surface area analysis using an automated machine having model NOVA 2200e of Quanta chrome manufacturer, USA. The X-ray diffraction analysis (XRD) of Fe3O4/C was performed using an X-ray Diffractometer (Joel, JDX-3532, Tokyo, Japan) having Ni filter, while mono-chromatic Cu-K$\alpha$ was taken as a source of radiation at operating wavelength of 1.5518 A°. The X-ray generator was operated at 40 kV and 30 mA current. The range and speed of scanning were 2 $\theta/\theta$ and 10 min$^{-1}$, respectively. The infrared spectra were obtained by analyzing the Fe$_3$O$_4$/C on the Fourier transform infrared spectrometer (FTIR; IR Prestige-21, Shimadzu, Kyoto, Japan) whereas the scanning range was from 4000 to 450 cm$^{-1}$. Topography of the Fe$_3$O$_4$/C was measured by placing the sample on the grid of the scanning electron microscope (SEM), where sputter coater (USA) was used for coating the sample with gold before its visualization under SEM. The images were recorded with a voltage of 20 KV (Joel JSM-5910). The Fe$_3$O$_4$/C was also analyzed by the thermogravimetric/thermal differential analysis (TG/DTA) through a Diamond series (USA) machine involving Al$_2$O$_3$ as a reference material. The Energy-dispersive X-ray spectroscopy (EDX) was executed for elemental analysis using an EDX model INCA 200, EDS X-sight apparatus (USA). Point of zero charge of Fe$_3$O$_4$/C was estimated using the mass titration method. In this method different amounts of Fe$_3$O$_4$/C were added a specific volume of double distilled water in nitrogen atmosphere with intermittent shaking and values of pH were noted after 24 h. The weights after being mixed were: 0.01%, 0.05%, 0.1%, 0.2%, 0.3%, 0.4% and 0.5% of total volume studied.

### 2.4. Batch Sorption Experiments

To avoid photodegradation of the CIPRO, the sorption experiments were carried out in 100 mL bottles. A stock solution having concentration 400 mg L$^{-1}$ was prepared in double distilled water. Through the dilution formula, various working solutions were prepared. Adsorption kinetics studies were performed on 40 and 80 mg L$^{-1}$ solutions whereas adsorption isothermal studies with 20, 40, 60, 80, 100 and 120 mg L$^{-1}$ solutions. In determining pH effect on adsorption, the pH of working solutions were adjusted with 0.1 M NaOH and 0.1 M HCl. Each bottle contained 0.04 g sorbents and 50 mL of the given concentration solutions that were shaken on a water-bath thermostatted mixer at moderate temperature. In kinetic studies samples were taken at pre-determined time intervals. The Fe$_3$O$_4$/C composite was removed from solution using a magnetic bar and then filtered through Whatman filter paper No. 1. The concentration of the CIPRO was determined with a UV/Visible spectrophotometer at 275 nm. All experiments including the blank tests were run in triplicate. The sorption percentage removal was calculated using following formulae:

$$\% \text{ removal} = \frac{C_o - C_t}{C_t} \times 100 \tag{1}$$

The amount adsorbed q$_t$ was calculated as:

$$q_t = \frac{C_o - C_t}{C_t} \times \frac{V}{m} \tag{2}$$

where initial concentration is presented as C$_o$ (mg L$^{-1}$), C$_t$ (mg L$^{-1}$) is concentration of selected antibiotic at time t, m (g) is the mass of the Fe$_3$O$_4$/C and V (L) is the volume of the solution taken.

It was significant to optimize the sorbent dosage as such type of analysis are required for the determination of the cost of the adsorption process. The sorbent contents were varied from 0.01 to 0.06 g to select optimum dosage of adsorbent for the subsequent experiments.

### 2.5. Equations and Model Used in the Study

The pseudo 1st order (Equation (3)) [19], 2nd order [20] (Equation (4)) and intraparticle diffusion [21] (Equation (5)) models were applied to describe the kinetics of selected antibiotic on prepared composite:

$$\ln (q_e - q_t) = \ln q_e - k_1 t \tag{3}$$

$$\frac{t}{q_t} = \frac{1}{k_2 q_{e2}} + \frac{1}{q_e} t \tag{4}$$

$$q_t = k_3 t^{1/2} + c \tag{5}$$

In the above equation $q_t$ and $q_e$ are the amount of the CIPRO in mg g$^{-1}$ adsorbed onto the surface of $Fe_3O_4/C$ at equilibrium and at times t (min), respectively. While, $k_1$ (min$^{-1}$), $k_2$ (g mg$^{-1}$ min$^{-1}$) and $k_3$ (mg g$^{-1}$ min$^{-0.5}$) are the adsorption's rate constants of the above-mentioned models.

Freundlich (Equation (7)), Langmuir (Equation (6)), Temkin (Equation (8)) and Jovanovich (Equation (9)) models [8,17,18] were used to evaluate the adsorption isotherm data.

$$\frac{C_e}{q_e} = \frac{1}{K_L q_{max}} + \frac{C_e}{q_{max}} \tag{6}$$

$$\ln q_e = \ln k_f + \ln \frac{C_e}{n} \tag{7}$$

$$q_e = \beta \ln \alpha + \beta \ln C_e \tag{8}$$

$$\ln q_e = \ln q_{max} + K_j C_e \tag{9}$$

In the above equations $C_e$ (mgL$^{-1}$) is the CIPRO equilibrium concentration in solution, $q_e$ is the amount of CIPRO adsorbed on to $Fe_3O_4/C$ at equilibrium, $q_{max}$ (mg g$^{-1}$) is the maximum adsorption capacity of $Fe_3O_4/C$, where $K_L$ is Langmuir constant, $k_f$ and $\frac{1}{n}$ are Freundlich constants, $\beta$ and $\alpha$ are Temkin constants, whereas $K_j$ is Jovanovich constant.

Van't Hoff equation (Equation (10)) was applied to determine values of different thermodynamic parameters, such as free energy, entropy and standard enthalpy [8,17,18]:

$$\ln k = \frac{\Delta S^\circ}{R} - \frac{\Delta H^\circ}{RT} \tag{10}$$

The Van't Hoff's plot; ln k vs 1/T, was used to estimate the value of k (Equation (11))

$$k = \frac{q_e}{c_e} \tag{11}$$

where k (L g$^{-1}$) is the coefficient of adsorption distribution, $q_e$ and $c_e$ are the amount of CIPRO adsorbed (mg g$^{-1}$) and remaining concentration in solution at equilibrium (mg L$^{-1}$) respectively. R (8.314 J mol$^{-1}$ K$^{-1}$) is a general gas constant and T (K) is absolute temperature.

The change in standard Gibbs free energy ($\Delta G^\circ$) was estimated using the following equation [8,17,18]:

$$\Delta G^\circ = \Delta H^\circ - T\Delta S^\circ \tag{12}$$

### 2.6. Membranes' Operation with and without Magnetic Composite

To recover the ciprofloxacin promptly from industrial effluents, RO, UF, and NF membrane mounted on a pilot plant were connected to a continuous stirred reactor where effect

of selected antibiotic as fouling agent was noted in form of decrease in permeate flux (J) and the percent retention (% R) by each membrane was also noted. All the membranes were first washed with distilled water according to the instruction of the manufacturer. A solution of known concentration of the CIPRO was prepared in distilled water. All samples were equilibrated to 298 K temperature, and pH 7.

Now in continuous stirred reactor $Fe_3O_4/C$ composite was added in same proportion as applied in adsorption experiment mentioned above. After the equilibration time adsorbent was collected by application magnet and effluents were now channelized into pilot plant where same amounts were passed from each membrane first from UF, then NF and finally from RO and the mentioned parameters in above paragraph were determined. The UF membraned was operated in a dead-end mode whereas the NF and RO were operated in cross end mode.

The % R of the CIPRO was determined by the following equation [8,17,18]:

$$\% R = \left[1 - \frac{C_P}{C_b}\right] \times 100 \tag{13}$$

In Equation (13), $C_P$ is the concentration (mg $L^{-1}$) of the CIPRO molecules in the permeate flux and $C_b$ is its concentration in mg$L^{-1}$ in bulk.

The permeate flux (J) (L $m^{-2}$ $h^{-1}$) was calculated at different time intervals during filtration through the formula [8,17,18]:

$$J = \frac{1}{A} \cdot \frac{dV}{dt} \tag{14}$$

In the above equation (Equation (14)), A is the area of the membrane (50 $m^2$ in case of the UF, and 3.2 $m^2$ in the case of the NF and the RO membranes), V is the volume of effluent (L) collected in a given interval and t is time interval (h). Membranes' backwashing was applied after successive experimental run.

## 3. Results and Discussion

### 3.1. Characterization of the $Fe_3O_4$/C Adsorbent

First, to confirm whether the resultant composite had a magnetic character or not, a bar magnet was brought near to the composite where all its particles adhered to magnet bar, thus, it was inferred that the composite had a magnetic character.

The EDX analyses of the $Fe_3O_4$/C composite is shown in Figure 2a. Carbon, iron, and oxygen are the major elements detected in the composite whereas some minute peaks of other elements are also there that may be impurities in the prepared sample. The major peaks observed confirms the formation of $Fe_3O_4$/C composite as these elements are the major elements in its formula [8,17,18,22].

The SEM images of the adsorbent with low and high magnifications are given in Figure 2b,c, providing a clear picture about the surface morphology of the $Fe_3O_4$/C composite. The composites have different shape and size particles. The presence of white particles in the images shows the existence of water of crystallization in the $Fe_3O_4$ cubic crystals present in the structure of $Fe_3O_4$/C composite. Aggregates in some places are visible in the pictures where particles have made clusters due to moisture contents [8,18,22].

XRD patterns of the $Fe_3O_4$/C is shown in Figure 2d. The peaks at 2θ value of 30.2°, 35°, 54.9° and 63° corresponded to 220, 311, 400, 511 and 440 indices, confirming the existence of the magnetite in the composite structure which needed for magnetic separation of the composite from the slurry after use [8,18,22,23]. The crystallite size of the $Fe_3O_4$/C composite was found to be 16 nm, calculated through Debye–Scherrer's equation [24].

$$D = K\frac{\lambda}{\beta} \cos\theta \tag{15}$$

where K is a constant and its value is equal to 0.9, λ (1.54 A°) is the wavelength of X-ray used, and β is the full width at half maximum of the peak in radians [25–27]. The average thickness of the nanocomposite was found to be 70 nm. Other parameters calculated from this equation are listed in Table 2 which have been estimated from plot 2e.

The FTIR spectra of the composite is given in Figure 2f. The spectra displayed the distinctive peaks with broad bands between 3500 and 3300 cm$^{-1}$, which may specify the existence of functional groups, such as $C_6H_5$-OH, -COOH and -COOH derivatives in addition to the existence of physically adsorbed water on the surface of the composite. The peak in the region of 3000–2800 cm$^{-1}$ corresponds to C-H alkanes, while two peaks in the region of 2000–1659 cm$^{-1}$ corresponds to the –CH-bond aromatic peak and peak at 1450 cm$^{-1}$ corresponds to the C = C aromatic bond, peaks at 1300–1000 cm$^{-1}$ corresponds to the -OH of alcohol and ether, while the peak at 575–580 cm$^{-1}$ corresponds to the Fe-O of magnetite and maghemite [8,18].

Figure 2g shows the TG/DTA pictogram of the sample. The TGA elucidated that the composite thermally quite stable as it resisted to mass loss to very high temperatures. At the early stage, at the temperature of 45–100 °C, a loss of 6.22% in total weight of the composite occurred which was attributed to dehydration of moisture present. At temperature 100–250 °C, another mass loss was experienced which is actually the loss water physically adsorbed and firmly bounded water to the surface of the composite [18,28]. The weight-loss continued up to 550 °C which was due to the decomposition of volatile organic matter as a result of combustion of carbon and phase transition of $Fe_3O_4$ to FeO, as FeO is thermodynamically stable above 570 °C [29]. Above 550 °C, the sample showed sufficient thermal stability and no further weight loss was observed. The final residue was carbon and a mixture of ash. In differential thermal analysis curve, there are three endothermic peaks in the temperature ranges from 40 °C to 450 °C.

The BET surface area and the BJH pore size distribution plots [30] of the $Fe_3O_4$/C composite are given in Figure 2h,i, respectively. The BET surface area determined was 51 m$^2$ g$^{-1}$ whereas from BJH pore size distribution plot the surface area of $Fe_3O_4$/C come out as 21.65 m$^2$ g$^{-1}$. The total pore volume and pore diameter of the adsorbent were 0.019 cm$^3$ g$^{-1}$ and 15.03 A°, respectively. In literature, attempts have been made to prepare magnetic adsorbents and efficient such adsorbents have been reported as well for example the micropore volume reported by Oliveira et al. [30] for different magnetic composites he prepared were in range 0.172 and 0.177 cm$^3$ g$^{-1}$ and in the findings of Anyika et al. [31] were in the range of 0.09 and 0.18 cm$^3$ g$^{-1}$.

The pH$_{pzc}$ (the pH at which the particle's net charge is zero or the particle possessed equal number of positive and negative charges) of the mango $Fe_3O_4$/C was found to be 7.3 (Figure 2j).

**Table 2.** Structural parameters of mango $Fe_3O_4$/C calculated from Debye–Scherrer equation.

| No | Parameters | Value |
|----|------------|-------|
| 1 | Crystallite size (D) | 0.8132 |
| 2 | Strain ($\varepsilon$) | $3.65 \times 10^{-3}$ |
| 3 | Full width at half maximum (FWHM) ($\beta$) | 0.010472 |
| 4 | Dislocation density ($\delta$) | $3.46 \times 10^{15}$ m$^{-2}$ |
| 5 | Number of crystallite per unit area (N) | $3.5 \times 10^{13}$ m$^{-2}$ |

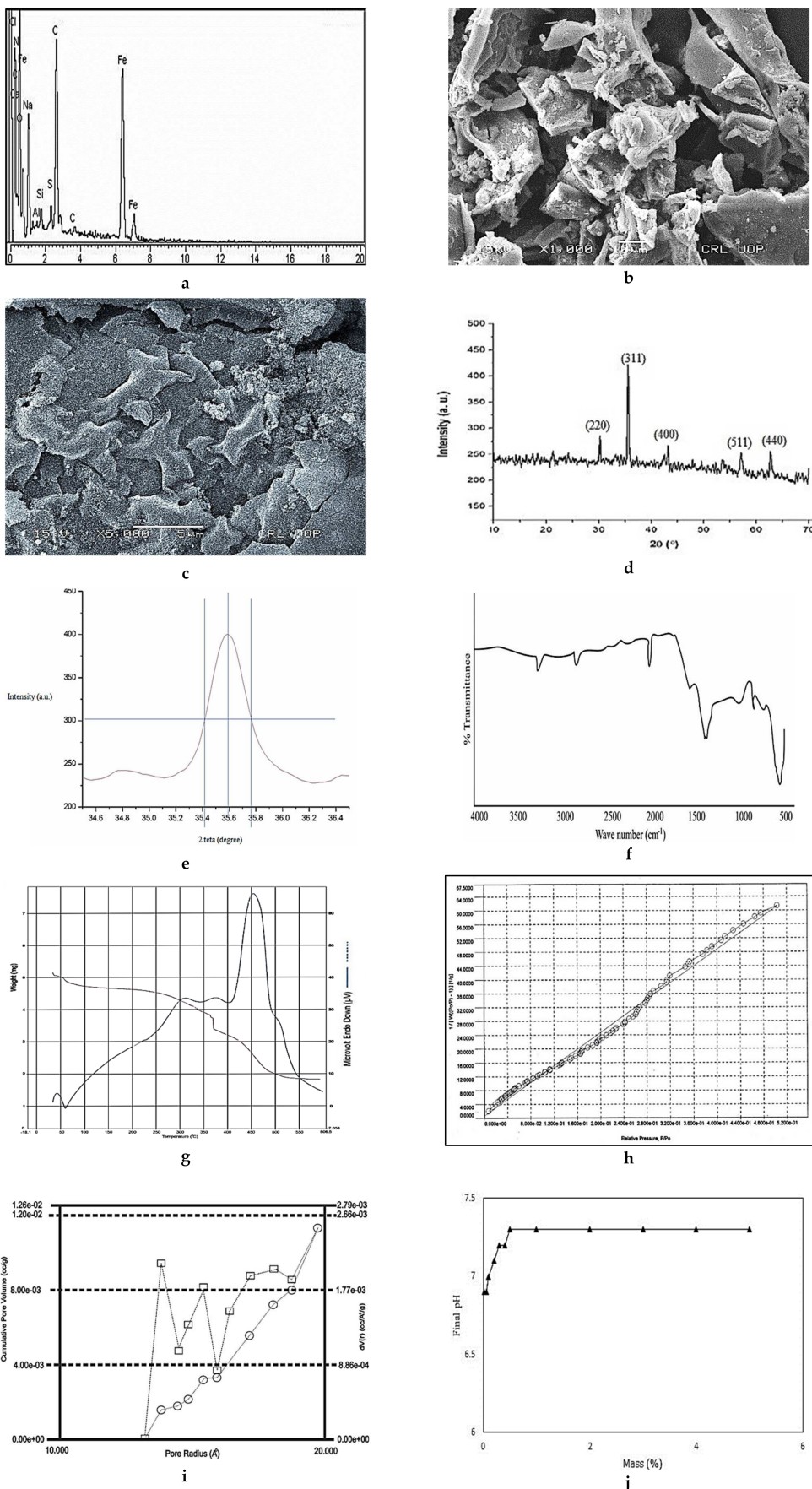

**Figure 2.** Characterization of Fe₃O₄/C (**a**) EDX (**b**) and (**c**) SEM (**d**) XRD (**e**) FTIR (**f**) Debye–Scherrer plot (**g**) TG/DTA (**h**) BET plot (**i**) BJH plot (**j**) pHpzc.

### 3.2. Isothermal and Kinetics Studies

The study of adsorption kinetics is widely used in deciding the effectiveness of an adsorbent, as such type of studies provide information about the mechanism of the process and optimum time to reach equilibrium, while taking a pollutant from an aqueous environment [8,17,18]. The effect of contact time is shown in Figure 3a, where fast adsorption of about 90% of the adsorbed CIPRO has taken place in initial half an hour. The resultants have coherence with the reported studies [17,18,32,33]. Fast initial adsorption is trailed by a slow stage where the adsorption rate has increased slowly and then flattened off after 1 h, pointing towards the slow diffusion of CIPRO into micropores. There was no significant alteration in the adsorption amount latter from 1 h and 2 h. Consequently, 1 h was taken as the equilibrium time of the CIPRO adsorption onto the $Fe_3O_4/C$ composite.

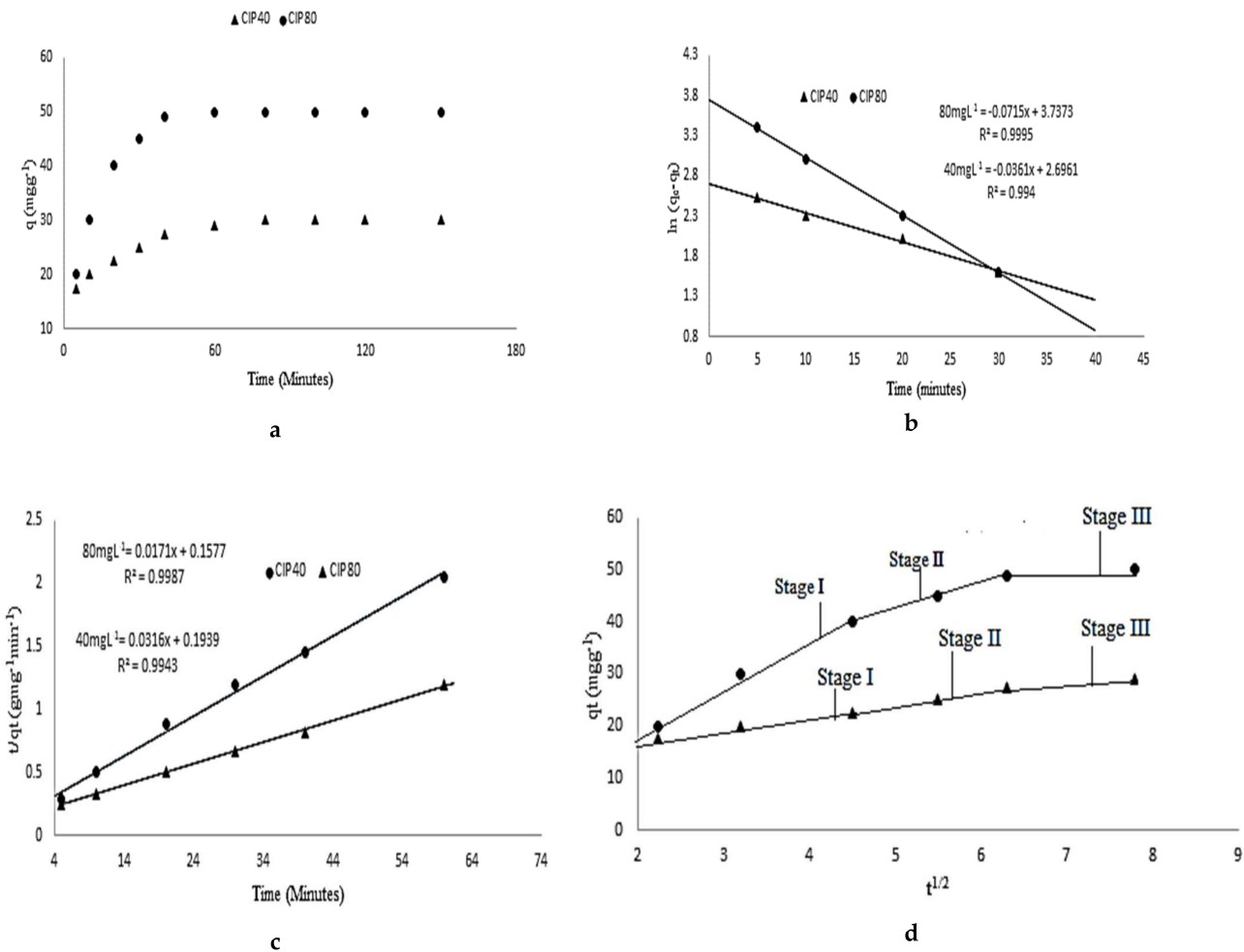

**Figure 3.** Adsorption kinetics plots of the CIPRO on the mango $Fe_3O_4/C$ composite: (**a**) effect of contact time on adsorption (**b**) Pseudo 1st order kinetics plot (**c**) Pseudo 2nd order kinetics plot (**d**) Intra particle diffusion plot.

The contact time effect data were fitted into three kinetic models (pseudo 1st order kinetics, pseudo 2nd order kinetics and the intraparticle diffusion) in order to gain the best fit model and consequently get the ultimate information about the adsorption process. Figure 3b,c shows the pseudo first and second order kinetics plots, respectively. Among these two the pseudo 2nd order kinetics model illustrates the adsorption kinetics well with $R^2$ value near to 1, (Table 3). The pseudo 2nd order kinetics model has been successful in elucidating chemical adsorption of contaminants by a given adsorbent from water where in such process sharing or exchange of electrons between the sorbent and sorbate are

predominant (covalent bonding and ion exchange) [34,35]. Thus, the main driving force in sorption may have come from these mentioned sharing or exchange of electrons between the CIPRO and the surface of the sorbents i.e., signifying the chemical nature of sorption process. Furthermore, the comparisons of the $q_{exp}$ with the $q_{cal}$ (from pseudo 1st and 2nd order kinetics models) also confirm that the pseudo 2nd order is most favorable model for explaining the process under study (Table 3).

The intraparticle diffusion plot is shown in Figure 3d. The plot shows that external surface sorption (stage 1st) took place within the initial five minutes, after that the 2nd stage designated as the slow adsorption phase starts, where intraparticle diffusion is rate restraining step. The 2nd stage is followed by a 3rd stage (a plateau stage) that can be accredited to the final equilibrium phase. The straight-line portion does not pass through the origin indicating that intraparticle diffusion is sole rate limiting step. The larger intercept values recommended that the process of sorption was mainly of exterior sorption. The values of the C and the k3 are given in the Table 3.

The isothermal adsorption parameters of the CIPRO adsorption onto the $Fe_3O_4/C$ composite, calculated from various models are given in Table 4. The Langmuir, Freundlich, Jovanovich and Temkin plots are shown in Figure 4a–d, respectively. From the $R^2$ value in the Table it is evident that the Langmuir model is well fitted ($R^2 = 0.997$). The maximum adsorption capability ($q_{max}$) was found to be 56.82 mg g$^{-1}$ as estimated from Langmuir model, followed by that estimated from Jovanovich model (15.33 mg g$^{-1}$). The higher adsorption capability of the $Fe_3O_4/C$ composite may be due to the nitrogen and oxygen containing functional groups, which plays a significant role in adsorption of different pollutants from water [36–39].

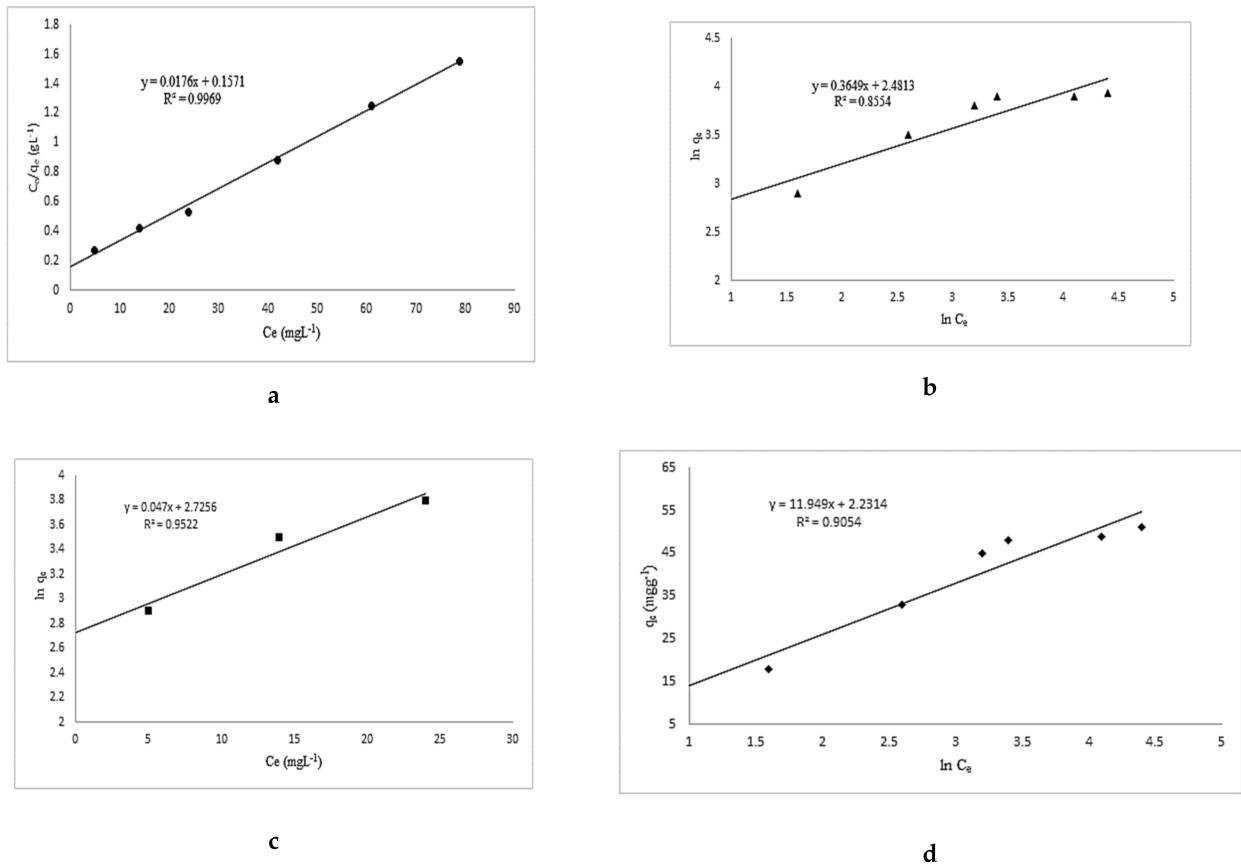

**Figure 4.** Adsorption isotherm plots of the CIPRO adsorption on the mango $Fe_3O_4/C$ composite. (**a**) Langmuir (**b**) Freundlich (**c**) Jovanovich isotherm (**d**) Temkin.

**Table 3.** Kinetic model parameters correlation co-efficient ($R^2$) for the CIPRO adsorption parameters on the mango $Fe_3O_4/C$ adsorbent.

| $C_o.$ (mg $L^{-1}$) | $q_{e\ exp}$ (mg $g^{-1}$) | Pseudo 1st Order k1 $R^2$ $q_{e\ cal}$ (mg $g^{-1}$) | Pseudo 1st Order k2 $R^2$ $q_e$ cal (mg $g^{-1}$) | Intra Particle Diffusion k3 $R^2$ C |
|---|---|---|---|---|
| 40 | 30 | 0.044 0.96 39.40 | 0.0410 0.993 30.10 | 2.3 0.91 3.10 |
| 80 | 50 | 0.051 0.96 93.00 | 0.0480 0.980 50.00 | 5.3 0.93 6.30 |

**Table 4.** Isotherm model parameters and correlation co-efficient ($R^2$) for CIPRO adsorption on $Fe_3O_4/C$.

| Langmuir | Freundlich | Temkin | Jovanovich |
|---|---|---|---|
| $q_{max}$ (mg $g^{-1}$) kL (L $mg^{-1}$) $R^2$ | KF 1/n $R^2$ | b β α $R^2$ | KJ $q_{max}$ (mg $g^{-1}$) $R^2$ |
| 56.82 0.112 0.997 | 12 0.37 0.86 | 207 11.95 1.2 0.91 | 0.047 15.33 0.95 |

### 3.3. The Effect of pH and Adsorbent Dosage on the Adsorption of the CIPRO Molecules

The influence of the pH on the adsorption of the selected antibiotic on $Fe_3O_4/C$ composite was evaluated in the pH range of 3–11. It is clear from Figure 5a, the CIPRO adsorption increases from pH 3 to 7. As in this pH range CIPRO existed as the cationic $CIPRO^+$, whereas at pH 7 it is in zwitterion form. When the pH of the solution became alkaline, a steady decrease in the CIPRO removal occurs, as anionic form ($CIPRO^-$) dominates. The point of zero charge values is also near 7 thus the positive interaction of adsorbent and adsorbate have been observed at pH 7 in the form of high adsorption values. The pH value of the media affects the surface charges of a given adsorbent and that of adsorbate leading to positive or negative interactions that is why best adsorption capacities are recorded at different pH for different adsorbents and adsorbates [40].

The effect of adsorbent dosage is shown in Figure 5b, where 0.05 g amount of adsorbent dosage was found optimum. Further increase in adsorbent dosage has caused little increase in percent removal of selected antibiotic. As the sorption capacity increased from 0.01 to 0.05 g is from 27% to 67% whereas for 0.06 g the increase in percent removal is comparatively small from that observed for 0.05 g dosage. The increase in % removal of the CIPRO might be due to the great number of available active sites in highest dose tested.

### 3.4. Effects of Hums (Humic Acid) on CIPRO Adsorption

Humic acid (HA) is a common natural organic matter found in water and often competes for active sites on adsorbent. HA molecules consist of –COOH, phenolic–OH and many other functional groups, which can interfere the interactions of the CIPRO molecules with the $Fe_3O_4/C$ composite. Therefore, it is of great significance to study the effect of the HA on the adsorption process. The effect of different concentrations of the HA (0–80 mg $L^{-1}$) on the adsorption of the CIPRO molecules from aqueous solution on the $Fe_3O_4/C$ was studied and as shown in Figure 5c. It is clear from the figure that the lower concentration of HA had a minor effect on the % removal of the CIPRO whereas the adsorption capacity has decreased with increasing the HA concentration. This mainly occurs because at low concentration, HA is adsorbed on the surface of the $Fe_3O_4/C$ by hydrogen bonding or by electrostatic interactions. When the HA concentration exceeds a certain limit, the number of free-moving humic acid molecules in the solution rises. The HA molecules form a soluble complex with CIPRO molecules, which blocks the pores on the surface of the $Fe_3O_4/C$, leading to a depression in the adsorption capacity [7,8].

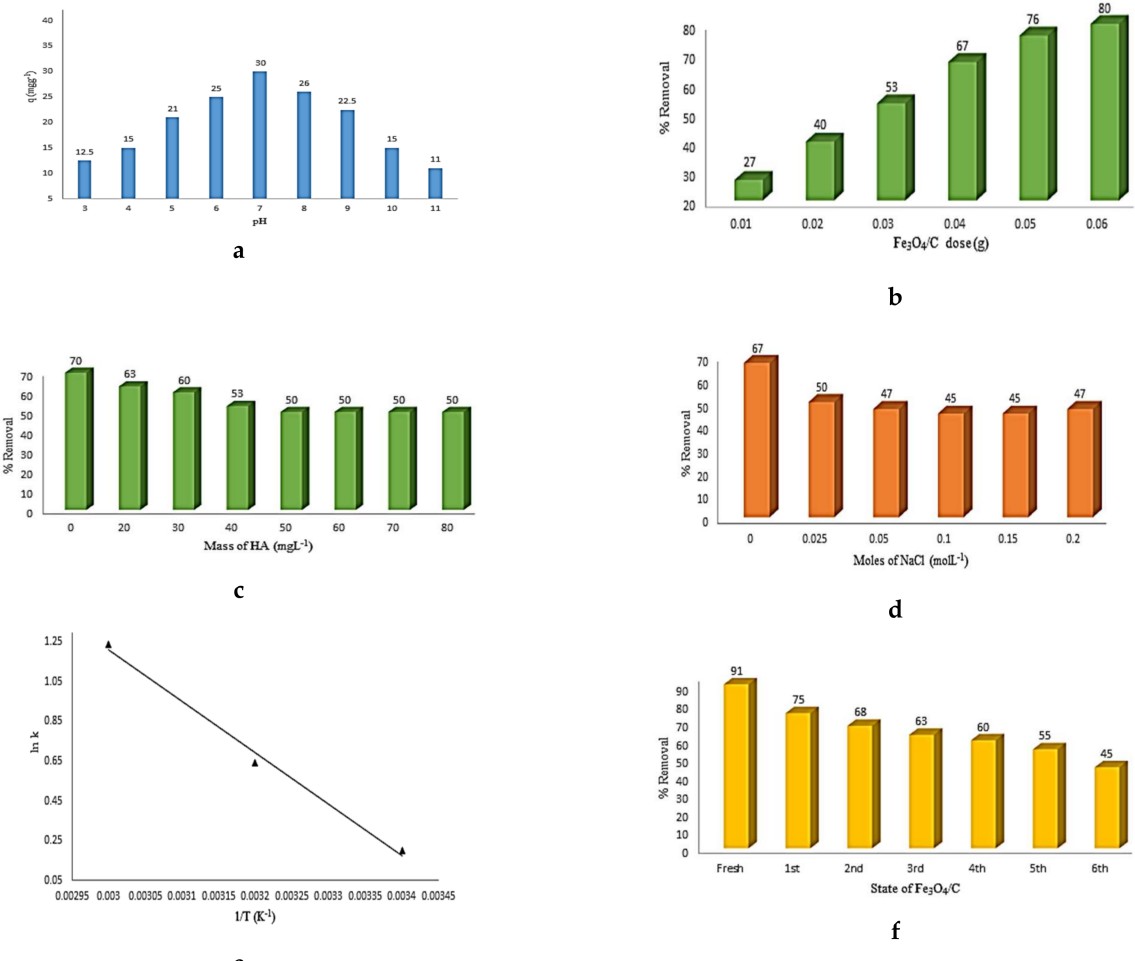

**Figure 5.** (**a**) Effect of pH on adsorption (**b**) Effect of adsorbent dosage. (**c**) Effect of Humic acid (**d**) effect of ionic strength (**e**) Van't Hoff plot effect of ionic strength. (**f**) Regeneration of the mango $Fe_3O_4/C$.

### 3.5. Effect of Ionic Strength (Sodium Chloride) on the CIPRO Adsorption

For the determination of the effect ionic strength on the CIPRO adsorption process, NaCl was used as ion source. The results of effect of ionic strength are given in Figure 5d. The results obtained showed that ionic strength had little effect on the CIPRO removal from solution. As the concentration of NaCl increased, both CIPRO and NaCl particles competes for the active site on the surface of $Fe_3O_4/C$. An increase in the concentration of NaCl weakened the interaction of the CIPRO particles with the $Fe_3O_4/C$ due to the compression of electrical double layer formed by $Na^+$ and $Cl^-$. Overall, it was concluded that NaCl solution had a little effect on the removal of the CIPRO molecules from aqueous solution [8].

### 3.6. Thermodynamics Parameters

The Van't Hoff is shown in Figure 5e whereas different thermodynamic parameters are shown in Table 5. From the values of the parameters in Table 5 it can be inferred that the process is feasible, spontaneous and exothermic in nature.

**Table 5.** Thermodynamic parameters for CIPRO adsorption on the mango $Fe_3O_4/C$.

| Temperature (K) | $\Delta H°$ (kJ mol$^{-1}$) | $\Delta S°$ (J mol$^{-1}$ K$^{-1}$) | $\Delta G°$ (kJmol$^{-1}$) |
|:---:|:---:|:---:|:---:|
| 298 | −21.6 | 75 | −2.40 |
| 313 | . . . .. | . . . .. | −2.60 |
| 333 | . . . .. | . . . .. | −2.73 |

### 3.7. Reusability/Regeneration and Recycling of Fe$_3$O$_4$/C

To further evaluate the regeneration and reusability of the $Fe_3O_4/C$ (Figure 5f), desorption experiments were carried out where 0.150 g of the $Fe_3O_4/C$ was added to 50 mL CIPRO solution having concentration of 80 mg L$^{-1}$ at the pH 7.0. The reaction mixture was mixed thoroughly at 25 °C in a water-bath for 6 h. The remaining concentration of the CIPRO in the filtrate was measured using a UV/Visible double beam spectrophotometer. The CIPRO loaded $Fe_3O_4/C$ was isolated from the reaction mixture with a magnet, and the solids were washed several times in a 3% NaOH solution, methanol and double distilled water [8]. Finally, the washed sample was oven-dried at 70 °C for 5 h. The same experiment was carried out six times under the same conditions and decrease in adsorption capacity was noted.

### 3.8. Membranes and Adsorption/Membrane Hybrid Processes

Fouling of membrane systems may be due to pore blocking, adsorption and cake formation. The concentration polarization effect of a pollutant near to membrane surfaces leading to fouling which affect the efficiency of membrane filtrations. The effects of concentration polarization are usually observed for a very short period of time at the initial stages of the membrane operations, and after this, the flux remains persistent in long-term applications. In order to check the effects of $Fe_3O_4/C$ on fouling caused by selected antibiotic, the membrane module was connected with a specially designed container equipped with an electromagnet in series where $Fe_3O_4/C$ was mixed with the CIPRO solution and stirred for approximately 1 h, as mentioned in material and method section.

The effect of selected antibiotic on the permeate flux of membranes (UF, NF and RO) are shown in Figure 6a–f while the % retention of CIPRO without and with prepared composite are given in Figure 7a–c. There is a clear decline in permeate flues of all the selected membranes even with double distilled water which most probably is due to the interaction of the ions present in double distilled water and may also be due to the intrinsic membrane resistance. The permeate flux of all membranes reached to a steady state after 20–30 min and was no longer changed within experimental cycle. The molecular weight of selected antibiotic (CIPRO = 331.346 g mol$^{-1}$) is smaller than molecular weight cutoff of UF membrane. Therefore, it was expected to pass freely from the UF membrane and the permeate concentration ($C_p$) should be the same as that of the concentration in the bulk ($C_b$) when membrane. However, still the concentration difference was more between $C_p$ and $C_b$ which might be due to adsorption of selected antibiotic on membrane surface that has affected the permeate fluxes as well. When adsorbent have been added the improvement in permeate fluxes and % retention is evident from figures.

The molecules of selected antibiotic were larger enough to be completely stopped from crossing the barriers by the NF and RO membranes. Here the % retention of selected antibiotics in both cases is almost 100% but the effect of concentration polarization in membrane fouling are quite drastic in comparison to UF membrane operation. Improved fluxes with adsorbent pretreatment in both cases are evident from the given figures.

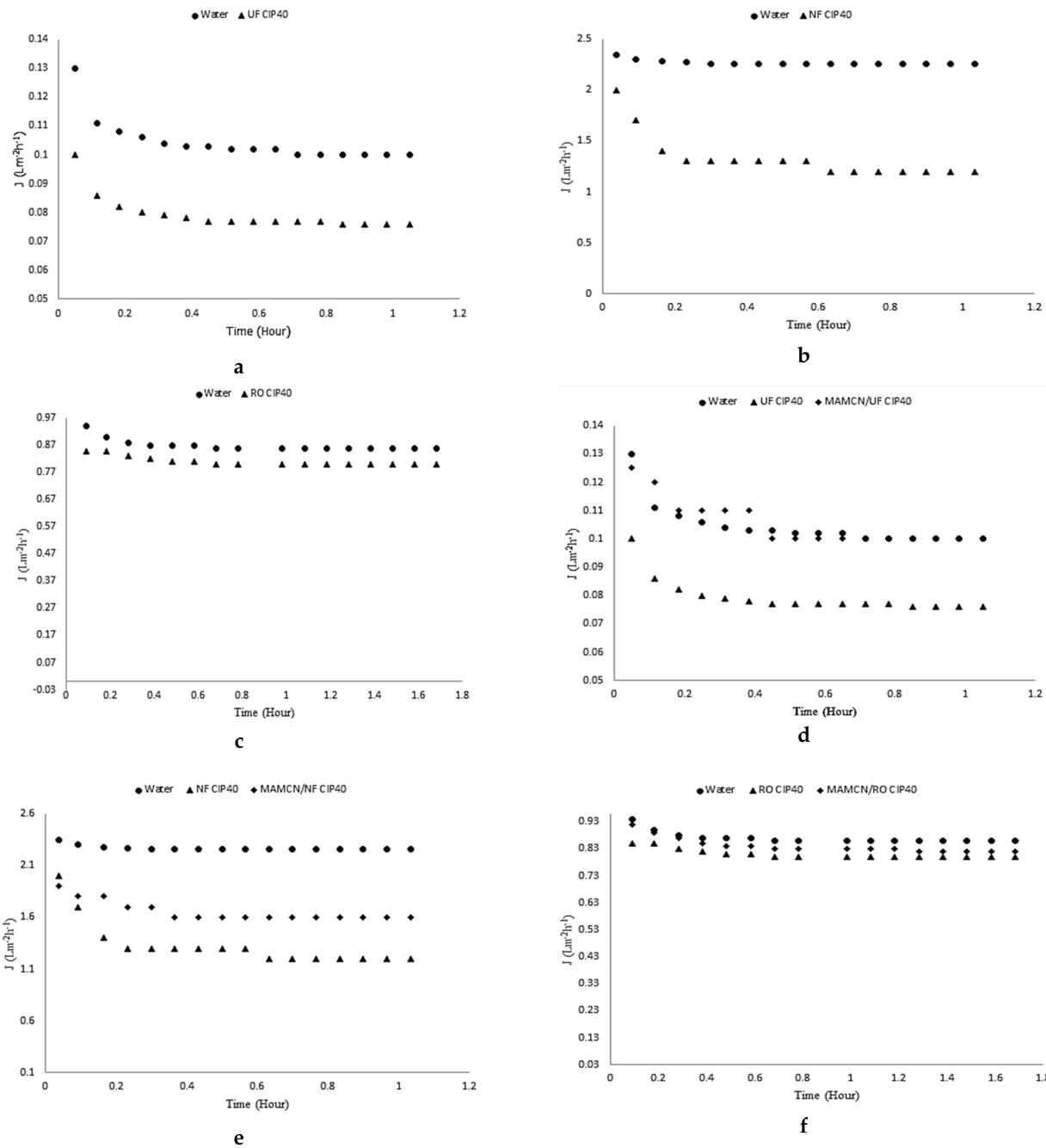

**Figure 6.** (**a**) Effect of CIPRO on permeate flux of UF membrane. (**b**) Effect of CIPRO on permeate flux of NF membrane. (**c**) Effect of CIPRO on permeate flux of RO membrane. (**d**) Improved permeate flux of UF membrane with mango $Fe_3O_4/C$. (**e**) Improved permeate flux of NF membrane with mango $Fe_3O_4/C$. (**f**) Improved permeate flux of RO membrane with mango $Fe_3O_4/C$.

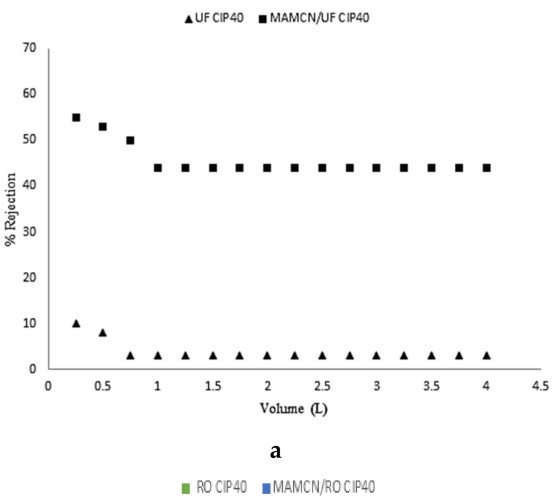

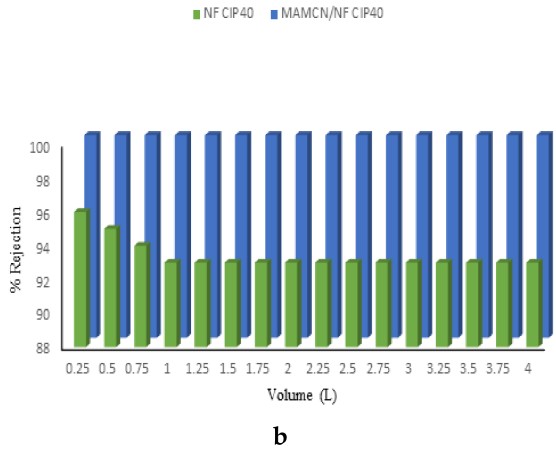

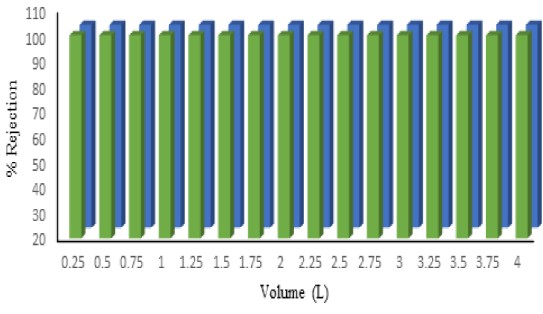

**Figure 7.** (**a**) Percent rejection of the CIPRO with UF membrane and $Fe_3O_4/C$ (**b**) Percent rejection of the CIPRO with NF membrane and the $Fe_3O_4/C/NF$ (**c**) Percent rejection of the CIPRO with the RO membrane and the $Fe_3O_4/C/RO$.

## 4. Conclusions

In this research work an attempt has been made to remove ciprofloxacin effectively from industrial effluents through a membrane pilot plant mounted on a steel stand with UF, NF and RO membranes. As a precautionary measure an iron-based magnetic carbon nanocomposites pretreatment was applied to minimize the chances of membrane fouling by the selected antibiotic. The $Fe_3O_4/C$ composite was prepared through coprecipitation method using mango peel as waste biomass. The intact iron oxide presence in the composite was mandatory, which was confirmed through FTIR and EDX techniques. The magnetic character of the composite was confirmed through XRD. Surface morphology was visualized through SEM whereas thermal stability was estimated from TG/DTA analysis. The point of zero charge and surface area were also determined. The prepared adsorbent was then effectively applied for the removal of ciprofloxacin from wastewater using batch adsorption approach. The equilibrium time was found to be 60 min, while kinetic data and isotherm data fitted well to pseudo 2nd order and Langmuir models, respectively. The thermodynamic aspects were also studied and from the results the spontaneity of the adsorption processes with exothermic nature was predicted. The prepared adsorbent was then used as pretreatment to control membrane fouling caused by selected antibiotic mounted on a specially designed pilot plant where improved percent retention and permeate fluxes were observed for the hybrid adsorption/membrane operations as compared to

naked membrane operations. Magnetic adsorbent effectively controlled the instant fouling in membranes caused by selected antibiotic and are thus recommended to be used as robust approach in controlling the antibiotics contamination in water reservoir that would thus consequently reduce the antibiotic resistance problem.

**Author Contributions:** Conceptualization, M.Z.; methodology, A.U.; software, S.A.; validation, I.Z., M.M. and M.Z.; formal analysis, A.S. and S.A.; investigation, M.Z. and I.Z.; resources, M.Z.; data curation, M.M.; writing—original draft preparation, M.Z., I.Z. and S.A.; writing—review and editing, I.Z.; visualization, M.M. and R.H.S.; supervision, M.Z.; project administration, M.Z.; funding acquisition, M.Z. All authors have read and agreed to the published version of the manuscript.

**Funding:** This research was funded by project SLTKT20427 "Sewage sludge treatment from heavy metals, emerging pollutants and recovery of metals by fungi" and by MLTKT19481R "Identifying best available technologies for decentralized wastewater treatment and resource recovery for India.

**Institutional Review Board Statement:** Not applicable.

**Informed Consent Statement:** Not applicable.

**Data Availability Statement:** The date presented in this study are available on request from the corresponding author.

**Conflicts of Interest:** The authors declare no conflict of interest. The funders had no role in the design of the study; in the collection, analyses, or interpretation of data; in the writing of the manuscript, or in the decision to publish the results.

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
