# Peer review of "Novel Magnetite Nanocomposites (Fe3O4/C) for Efficient Immobilization of Ciprofloxacin from Aqueous Solutions through Adsorption Pretreatment and Membrane Processes"

_water, doi:10.3390/w14050724_

Round 1

Reviewer 1 Report

This manuscript cannot be recommended for acceptance because of the following reasons.

  1. This work lacks novelty. A lot of similar manuscripts has been published recently. These are some examples below.

- Zhou, Yue, Shurui Cao, Cunxian Xi, Xianliang Li, Lei Zhang, Guomin Wang, and Zhiqiong Chen. "A novel Fe3O4/graphene oxide/citrus peel-derived bio-char based nanocomposite with enhanced adsorption affinity and sensitivity of ciprofloxacin and sparfloxacin." Bioresource technology 292 (2019): 121951.

- Mahmoud, Mohamed E., Shaimaa R. Saad, Abdel Moneim El-Ghanam, and Rabah Hanem A. Mohamed. "Developed magnetic Fe3O4–MoO3-AC nanocomposite for effective removal of ciprofloxacin from water." Materials Chemistry and Physics 257 (2021): 123454.

- Wang, Yue, Xiaoxiao Zhu, Dongqing Feng, Anthony K. Hodge, Liujiang Hu, Jinhong Lü, and Jianfa Li. "Biochar-supported FeS/Fe3O4 composite for catalyzed Fenton-type degradation of ciprofloxacin." Catalysts 9, no. 12 (2019): 1062.

- Balarak, Davoud, Mohadeseh Zafariyan, and Kethineni Chandrika. "Adsorption of ciprofloxacin from aqueous solution onto Fe3O4/graphene oxide nanocomposite." International Journal of Pharmaceutical Sciences and Research 11, no. 1 (2020): 268-274.

- Palacio, Daniel A., Bernabe L. Rivas, and Bruno F. Urbano. "Ultrafiltration membranes with three water-soluble polyelectrolyte copolymers to remove ciprofloxacin from aqueous systems." Chemical Engineering Journal 351 (2018): 85-93.

  1. Authors do not understand the concept of crosslinking in membrane technology. Crosslinking provides covalent linkages between membrane components. The context in which crosslinking is used in the manuscript is inappropriate. Authors should rather use “post-treatment”.

Author Response

Reviewer 1

This manuscript cannot be recommended for acceptance because of the following reasons.

  1. This work lacks novelty. A lot of similar manuscripts has been published recently. These are some examples below.
  • Worthy reviewer, the work is novel and In the following lines I will prove that our work is not similar to those you quoted in form of published papers.

- Zhou, Yue, Shurui Cao, Cunxian Xi, Xianliang Li, Lei Zhang, Guomin Wang, and Zhiqiong Chen. "A novel Fe3O4/graphene oxide/citrus peel-derived bio-char based nanocomposite with enhanced adsorption affinity and sensitivity of ciprofloxacin and sparfloxacin." Bioresource technology 292 (2019): 121951.

  • This paper contain graphene where we do not have used graphene. Where are membranes studies in this paper? On which basis you can justify your claim.

- Mahmoud, Mohamed E., Shaimaa R. Saad, Abdel Moneim El-Ghanam, and Rabah Hanem A. Mohamed. "Developed magnetic Fe3O4–MoO3-AC nanocomposite for effective removal of ciprofloxacin from water." Materials Chemistry and Physics 257 (2021): 123454.

  • Here the composite is a ternary complex and membrane studies are totally absent.

- Wang, Yue, Xiaoxiao Zhu, Dongqing Feng, Anthony K. Hodge, Liujiang Hu, Jinhong Lü, and Jianfa Li. "Biochar-supported FeS/Fe3O4 composite for catalyzed Fenton-type degradation of ciprofloxacin." Catalysts 9, no. 12 (2019): 1062.

  • This study is about degradation where my study is about adsorption. Also composite is totally different. No membrane studies are involved.

- Balarak, Davoud, Mohadeseh Zafariyan, and Kethineni Chandrika. "Adsorption of ciprofloxacin from aqueous solution onto Fe3O4/graphene oxide nanocomposite." International Journal of Pharmaceutical Sciences and Research 11, no. 1 (2020): 268-274.

  • Here graphene composites have been used which are quite different from our composite and membrane studies are also absent.

- Palacio, Daniel A., Bernabe L. Rivas, and Bruno F. Urbano. "Ultrafiltration membranes with three water-soluble polyelectrolyte copolymers to remove ciprofloxacin from aqueous systems." Chemical Engineering Journal 351 (2018): 85-93.

  • Here the only membrane involve is UF where in my study two other membranes NF and RO have been used, Here, soluble polyelectrolyte copolymers have been used as adsorbent but in my study there is no polymer involve 
  1. Authors do not understand the concept of crosslinking in membrane technology. Crosslinking provides covalent linkages between membrane components. The context in which crosslinking is used in the manuscript is inappropriate. Authors should rather use “post-treatment”.
  • Worthy reviewer, the title was modified as: Novel magnetite nanocomposites (Fe3O4/C) for efficient immobilization of Ciprofloxacin from aqueous solutions through adsorption pretreatment and membrane processes

Reviewer 2 Report

Major comments:

The study comprised of “Novel magnetite nanocomposites (Fe3O4/C) for efficient immo-2 bilization of Ciprofloxacin from aqueous solutions through ad-3 sorption and membrane cross-linking processes”. The study was designed to fabricate iron-based magnetic carbon nanocomposites (Fe3O4/C) from mango biomass precursors and utilized as an adsorbent for the ciprofloxacin removal from wastewater through batch adsorption and adsorption/membrane cross-linking processes. Although the study contains many findings, it has some important shortcomings that need to be addressed. One of them, introduction section; the lack of proper linkage between the different paragraphs. In addition to that, there is the repetition of ideas at several points, there is also the confusion between material and methods and results and discussion. The literature review is not strong enough to provide research gaps. The results and discussion section is the most important part of a study, but no critical discussion has been made. As this section progress, there is lacking of connection of results with each other. The English language used in the study needs improvement, as there are punctuation and grammatical mistakes throughout the manuscript. Sentences need more clarity and better construction. Study recent literature and emphasis the latest topics and trends in this region. It is obvious the quality of the manuscript does not meet the standards of the Water Journal, therefore, needs major revisions or should be rejected in its present form. The authors are advised to address the following comments very carefully.

Introduction:

The introduction needs to be more strengthened in terms of recent research in this area with possible research gaps and future applications of this study that eventually led to the lack of novelty. There is lack of sufficient background information, which is unable to give the reader detailed background knowledge. The introduction needs to be more emphasized on the research work with a detailed explanation of the whole process considering past, present and future scope. It is strongly recommended to add a recent literature survey about various types of wastewater treatment plants, various technologies to reduce environmental pollution and how these technologies fuels affect the current pollution levels and alarming global warming. Research gaps should be highlighted more clearly and future applications of this study should be added.

Specific Comments:

  1. The title is not appropriate, authors are advised to revise the tile, which should be comprehensive and novel.
  2. Keywords: Please add specific and novel keywords.
  3. Page 2, line 46-47-48-49: “The occurrence of antibiotics……. modification of the algae’s structure in aquatic habitats and intervention with the plants’s photosynthesis”. Add a scoping value field that supports that pharmaceutical residues have an impact on the environment?
  4. Page 1-4: Introduction and background is weak, no strong information is provided about the different types of environmental/water pollutions their effects on humans such as from wastewater industries the major contributor in environmental pollution, therefore, the authors are advised to read and add environmental pollution types and levels from the following studies: Environmental Science and Pollution Research, 2021; 28, 48505–48516. Green Technologies for the Defluoridation of Water, 2021; Jan 1 (pp. 41-88), Elsevier https://doi.org/10.1016/B978-0-323-85768-0.00004-X. Chemosphere, 2021;282:131056. Journal of Molecular Liquids, 2021; 327:114791. Chemosphere, 2022;287:132319.
  5. Page 2, line 57-58-59-60: “The CIPRO has been detected from the aquatic environment in the concentration ranges from mg L-1 to ng L-1”. “Although, the CIPRO concentration in the aquatic environment …. potential threats 60 to aquatic organisms”. This paragraph needs to be rewritten.
  6. Page 2, line 89: “To synthesize iron-based carbon nanocomposites (Fe3O4/C) from mango biomass precursors...”. Why did the authors choose mango biomass precursors to synthesize (Fe3O4/C)?
  7. Page 3, Section Preparation of Mango Magnetic Carbon Nanocomposites (Fe3O4/C): The authors did not mention which standard methods were followed.
  8. Page 3, section 2.3 Textural characterizations: Authors have not mentioned what standard methods have been followed, please compare SEM/XRD/FTIR techniques and molecular models with the following recent studies: Materials Science and Engineering: C, 2021;126:112127. Journal of the Taiwan Institute of Chemical Engineers, 2021;125:141-152. Coatings, 2021;11(4):386. Processes, 2021;9(9):1610.
  9. Page 4, section Batch sorption experiments: “To avoid photodegradation of the CIPRO, the sorption tests were accompanied in 100 140 mL glass bottles with a brown color.” In the introduction section you mentioned that CIPRO has a long half-life and low biodegradability and now to avoid photodegradation of CIPRO..., how is this possible? The two ideas are contradictory. please explain.
  10. Page 4, section Batch sorption experiments: why didn't the authors insert the remaining concentration of CIPRO into the solution? is not necessary? Please explain.
  11. Most of the mathematical/reaction equations are not cross-referencing in the text. Please cross-reference each equation properly in the text throughout the manuscript.
  12. Page 4, section Effect of Solid Content: “Fig. 6b showed that by elevation in the solid con-160 tent (sorbent dose) from 0.01 to 0.04 g..”: Please do not give the result, just put the essential things.
  13. Page 6, section Results and Discussion- Parameters of the Mango Fe3O4/C Adsorbent: The EDX analyses of the mango Fe3O4/C adsorbent shown in figure 3a needs special attention, the elements are on chovement. Please compare your results with the following recent studies: Hindawi Journal of Chemistry Volume 2021, Article ID 6678588 and Hindawi Adsorption Science & Technology Volume 2021, Article ID 2359110.
  14. More recent research about types of water technologies, wastewater pollution, treatment methods and pollution reduction technologies is suggested to be added to make the background and discussion more strong: Case Studies in Chemical and Environmental Engineering, 2021;3:100083. Environmental Research, 2022;204:112387. Chemosphere, 2022;288:132606. Chemosphere, 2022;289:133222. Journal of King Saud University-Science, 2022;34(2):101745. Chemosphere, 2022;287:132453. Environmental Science and Pollution Research, 2021 Sep 24:1-9. https://doi.org/10.1007/s11356-021-16390-0. Scientific Reports, 2021;11(1): 17952.
  15. Page 9, section Adsorption Studies of the Mango Fe3O4/C: “As shown in Figure 4a, the kinetics curve for the CIPRO was composed of a fast preliminary adsorption where approximately 90% of the CIPRO was adsorbed within half an hour.” In figure 3a, the authors showed that 90% of the CIPRO was adsorbed. On the other hand, in figure 3b, we saw that 50% of the CIPRO in two concentrations (CIP40 and CIP80) still remained in the solution. how can explain that?
  16. Conclusion: The conclusions only discuss a few of the researched criteria, which is insufficient to portray the full picture of this study's contribution. The authors should offer detailed conclusions that include key values, the applicability of the applied approach, major discoveries, contributions, and possible future work.
  17. The graphs throughout the manuscript are not consistent, some coloured, some black and white with blurry resolution. Revise all graphs with high-quality images and keep all figures as coloured with consistent fonts. The units stated in the graphs axis need to be double-checked.
  18. The authors are advised to revise references, including the latest references. Please see some suggestions in the specific comments and for the ‘introduction’ section.

Author Response

Reviewer 2

The study comprised of “Novel magnetite nanocomposites (Fe3O4/C) for efficient immo-2 bilization of Ciprofloxacin from aqueous solutions through ad-3 sorption and membrane cross-linking processes”. The study was designed to fabricate iron-based magnetic carbon nanocomposites (Fe3O4/C) from mango biomass precursors and utilized as an adsorbent for the ciprofloxacin removal from wastewater through batch adsorption and adsorption/membrane cross-linking processes. Although the study contains many findings, it has some important shortcomings that need to be addressed.

  • Thank you worthy reviewer, for the encouraging remarks.

One of them, introduction section; the lack of proper linkage between the different paragraphs. In addition to that, there is the repetition of ideas at several points, there is also the confusion between material and methods and results and discussion.

  • Introduction was revised in meaningful way. The whole section has been rephrased and interlinked, highlighted as yellow.

The literature review is not strong enough to provide research gaps.

  • The literature review was modified in such a way to cover the gaps

The results and discussion section is the most important part of a study, but no critical discussion has been made. As this section progress, there is lacking of connection of results with each other.

  • The whole section has been revised. Extra discussion has been omitted and only critical discussion has been made over there instead.

The English language used in the study needs improvement, as there are punctuation and grammatical mistakes throughout the manuscript.

  • The language was improved accordingly.

Sentences need more clarity and better construction. Study recent literature and emphasis the latest topics and trends in this region. It is obvious the quality of the manuscript does not meet the standards of the Water Journal, therefore, needs major revisions or should be rejected in its present form. The authors are advised to address the following comments very carefully.

  • Worthy reviewer, thank you very much for your positive input with critically reading the scripts. We have tried our best to resolve all the issues you pointed out. Hopefully, it will be ok now.

Introduction:

The introduction needs to be more strengthened in terms of recent research in this area with possible research gaps and future applications of this study that eventually led to the lack of novelty. There is lack of sufficient background information, which is unable to give the reader detailed background knowledge. The introduction needs to be more emphasized on the research work with a detailed explanation of the whole process considering past, present and future scope. It is strongly recommended to add a recent literature survey about various types of wastewater treatment plants, various technologies to reduce environmental pollution and how these technologies fuels affect the current pollution levels and alarming global warming. Research gaps should be highlighted more clearly and future applications of this study should be added.

  • Worthy reviewer, this whole section has been revised in such way to incorporate all your worthy suggestions along with future perspectives. Hope it will be ok now.

Specific Comments:

  1. The title is not appropriate, authors are advised to revise the tile, which should be comprehensive and novel.
  • Worthy reviewer, it has been revised accordingly.
  1. Keywords: Please add specific and novel keywords.
  • Thank you worthy reviewer, they were added accordingly.
  1. Page 2, line 46-47-48-49: “The occurrence of antibiotics……. modification of the algae’s structure in aquatic habitats and intervention with the plants’s photosynthesis”. Add a scoping value field that supports that pharmaceutical residues have an impact on the environment?
  • Worthy reviewer, the added statement was not appropriate it was rephrased accordingly with appropriate references.
  1. Page 1-4: Introduction and background is weak, no strong information is provided about the different types of environmental/water pollutions their effects on humans such as from wastewater industries the major contributor in environmental pollution, therefore, the authors are advised to read and add environmental pollution types and levels from the following studies: Environmental Science and Pollution Research, 2021; 28, 48505–48516. Green Technologies for the Defluoridation of Water, 2021; Jan 1 (pp. 41-88), Elsevier https://doi.org/10.1016/B978-0-323-85768-0.00004-X. Chemosphere, 2021;282:131056. Journal of Molecular Liquids, 2021; 327:114791. Chemosphere, 2022;287:132319.
  • Thank you, worthy reviewer, none of these studies are relevant to my work. However, one in the remaining suggested references is relevant and it has been cited as reference no 19
  1. Page 2, line 57-58-59-60: “The CIPRO has been detected from the aquatic environment in the concentration ranges from mg L-1 to ng L-1”. “Although, the CIPRO concentration in the aquatic environment …. potential threats 60 to aquatic organisms”. This paragraph needs to be rewritten.
  • The paragraph was rephrased accordingly.
  1. Page 2, line 89: “To synthesize iron-based carbon nanocomposites (Fe3O4/C) from mango biomass precursors...”. Why did the authors choose mango biomass precursors to synthesize (Fe3O4/C)?
  • Worthy reviewer, as mango are abundantly found in Pakistan and heaps of its peel in summer are every where in residential areas where in rainy season give rise to unpleasant smell. As a solution to use it for beneficial purpose it has been used apart from the fact that has high cellulose fibrous contents.
  1. Page 3, Section Preparation of Mango Magnetic Carbon Nanocomposites (Fe3O4/C): The authors did not mention which standard methods were followed.
  • Worthy reviewer, the coprecipitation method has been used which has been mentioned accordingly there.
  1. Page 3, section 2.3 Textural characterizations: Authors have not mentioned what standard methods have been followed, please compare SEM/XRD/FTIR techniques and molecular models with the following recent studies: Materials Science and Engineering: C, 2021;126:112127. Journal of the Taiwan Institute of Chemical Engineers, 2021;125:141-152. Coatings, 2021;11(4):386. Processes, 2021;9(9):1610.
  • Worthy reviewer, the detailed methods are already there. References are not mostly needed in this part. Also it is my own devised method.
  1. Page 4, section Batch sorption experiments: “To avoid photodegradation of the CIPRO, the sorption tests were accompanied in 100 140 mL glass bottles with a brown color.” In the introduction section you mentioned that CIPRO has a long half-life and low biodegradability and now to avoid photodegradation of CIPRO..., how is this possible? The two ideas are contradictory. please explain.
  • Worthy reviewer, the brown color bottle word was omitted accordingly. Thanks for the correction.
  1. Page 4, section Batch sorption experiments: why didn't the authors insert the remaining concentration of CIPRO into the solution? is not necessary? Please explain.
  • Worthy reviewer, there is some mistake in your sentence that is why the sense is not clear. However, as far I guess you are taking about solution concentration that were mentioned accordingly there in the specified section.
  1. Most of the mathematical/reaction equations are not cross-referencing in the text. Please cross-reference each equation properly in the text throughout the manuscript.
  • Appropriate references were cited accordingly there and numbering of the equations were also revised.
  1. Page 4, section Effect of Solid Content: “Fig. 6b showed that by elevation in the solid con-160 tent (sorbent dose) from 0.01 to 0.04 g..”: Please do not give the result, just put the essential things.
  • Worthy reviewer, the section has been revised accordingly. The sentences related to discussion were moved to discussion section accordingly.
  1. Page 6, section Results and Discussion- Parameters of the Mango Fe3O4/C Adsorbent: The EDX analyses of the mango Fe3O4/C adsorbent shown in figure 3a needs special attention, the elements are on chovement. Please compare your results with the following recent studies: Hindawi Journal of Chemistry Volume 2021, Article ID 6678588 and Hindawi Adsorption Science & Technology Volume 2021, Article ID 2359110.
  • Relevant references are already there and you are describing a spectra the facts are in graph that do not needs citations. However, one of them is relevant of to my study was added in introduction section.
  1. More recent research about types of water technologies, wastewater pollution, treatment methods and pollution reduction technologies is suggested to be added to make the background and discussion more strong: Case Studies in Chemical and Environmental Engineering, 2021;3:100083. Environmental Research, 2022;204:112387. Chemosphere, 2022;288:132606. Chemosphere, 2022;289:133222. Journal of King Saud University-Science, 2022;34(2):101745. Chemosphere, 2022;287:132453. Environmental Science and Pollution Research, 2021 Sep 24:1-9. https://doi.org/10.1007/s11356-021-16390-0. Scientific Reports, 2021;11(1): 17952.
  • Worthy reviewer, there are already more than 40 references and more than 40 you have suggested which would make the research paper as a review paper.
  1. Page 9, section Adsorption Studies of the Mango Fe3O4/C: “As shown in Figure 4a, the kinetics curve for the CIPRO was composed of a fast preliminary adsorption where approximately 90% of the CIPRO was adsorbed within half an hour.” In figure 3a, the authors showed that 90% of the CIPRO was adsorbed. On the other hand, in figure 3b, we saw that 50% of the CIPRO in two concentrations (CIP40 and CIP80) still remained in the solution. how can explain that?
  • Worthy reviewer, this sentence was revised accordingly and the sense is about the adsorbed amount not of overall concentration used. It mean out of the total adsorbed amount 90% has been adsorbed in 30 min.
  1. Conclusion: The conclusions only discuss a few of the researched criteria, which is insufficient to portray the full picture of this study's contribution. The authors should offer detailed conclusions that include key values, the applicability of the applied approach, major discoveries, contributions, and possible future work.
  • Conclusion section was revised accordingly.
  1. The graphs throughout the manuscript are not consistent, some coloured, some black and white with blurry resolution. Revise all graphs with high-quality images and keep all figures as coloured with consistent fonts. The units stated in the graphs axis need to be double-checked.
  • Worthy reviewer, different software produces different types of effects in the drawn graphs or picture which we cannot change. However, resolution of the pictures was increased accordingly.
  1. The authors are advised to revise references, including the latest references. Please see some suggestions in the specific comments and for the ‘introduction’ section.
  • Worthy reviewer, appropriate from them have been added accordingly.

Round 2

Reviewer 1 Report

The revised version is ok.

Author Response

Reviewer 1:

The revised version is ok.

Thank you, worthy reviewer, for the positive remarks.

Reviewer 2 Report

Major comments:

The study comprised of “Novel magnetite nanocomposites (Fe3O4/C) for efficient immo-2 bilization of Ciprofloxacin from aqueous solutions through ad-3 sorption and membrane cross-linking processes”. The study was designed to fabricate iron-based magnetic carbon nanocomposites (Fe3O4/C) from mango biomass precursors and utilized as an adsorbent for the ciprofloxacin removal from wastewater through batch adsorption and adsorption/membrane cross-linking processes.

  1. The results shown in this study are lack of innovation and novelty. One of the major flaws is that the experimental design and results are rather simple, making the manuscript lack of strong elucidation to support their novelty.
  2. The literature review is still poor and not strong enough to provide research gaps to prove the novelty of this work. The background of the study is missing, no comparison has been made with the previous data and other renewable energy sources. The identification of the research gaps is still missing.
  3. Furthermore, the results and discussion section still lacks with proper discussion in comparison to the previous studies.
  4. The authors have failed in explaining the results with proper discussion and did not clearly discuss the reasons for trends reported in the manuscript.
  5. Some of the figures and tables are just inserted without proper explanation which is insufficient to present the complete pictures of the present research study.
  6. There is a repetition of sentences having the same concept present throughout the manuscript.
  7. The English language used in the manuscript still needs major improvements as there are many punctuation and grammatical mistakes present throughout the manuscript. Sentences need more clarity and better construction.
  8. The authors have ignored and deleted some of the reviewer’s comments without bothering to address and make the required changes in the revision.
  9. The methodology section is still very poor.

In short, there is a lack of convincing evidence to support their novelty in the manuscript. It is obvious the quality of the manuscript does not meet the standards of the water Journal therefore, should be rejected.

Author Response

Reviewer 2:

The study comprised of “Novel magnetite nanocomposites (Fe3O4/C) for efficient immo-2 bilization of Ciprofloxacin from aqueous solutions through ad-3 sorption and membrane cross-linking processes”. The study was designed to fabricate iron-based magnetic carbon nanocomposites (Fe3O4/C) from mango biomass precursors and utilized as an adsorbent for the ciprofloxacin removal from wastewater through batch adsorption and adsorption/membrane cross-linking processes.

  1. The results shown in this study are lack of innovation and novelty. One of the major flaws is that the experimental design and results are rather simple, making the manuscript lack of strong elucidation to support their novelty.
  • Worthy reviewer, do you think if experimental design and results are simple will not be novel. Please quote this rue where it has been stated.
  1. The literature review is still poor and not strong enough to provide research gaps to prove the novelty of this work. The background of the study is missing, no comparison has been made with the previous data and other renewable energy sources. The identification of the research gaps is still missing.
  • Worthy reviewer, first in this study we are dealing with membrane fouling in this study not with renewable energy sources then why I should make a comparison with non-relevant thing. For information worthy reviewer, please note it that I am the only author working to control membrane fouling through magnetic adsorbents. If google it you will only find my papers. However, as you know it better self-citation in most journals beyond from certain limit are not allowed.
  1. Furthermore, the results and discussion section still lacks with proper discussion in comparison to the previous studies.
  • Worthy reviewer, as I stated before on this topic if you google it you will only find y name then how can I made comparison with other studies. The study is not about renewable energy sources to compare with that.
  1. The authors have failed in explaining the results with proper discussion and did not clearly discuss the reasons for trends reported in the manuscript.
  • Worthy reviewer please consult my response to point 3 and 4.
  1. Some of the figures and tables are just inserted without proper explanation which is insufficient to present the complete pictures of the present research study.
  • Worthy reviewer, they were again checked and all required details were found there.
  1. There is a repetition of sentences having the same concept present throughout the manuscript.
  • Worthy reviewer, in initial review there were such repetitions as it was written by student but being expert in my field with more than 170 publications, I have revised by myself. Please tell me about the line numbers where these repetitions are.
  1. The English language used in the manuscript still needs major improvements as there are many punctuation and grammatical mistakes present throughout the manuscript. Sentences need more clarity and better construction.
  • The language was rechecked and corrected where needed.
  1. The authors have ignored and deleted some of the reviewer’s comments without bothering to address and make the required changes in the revision.
  • Worthy reviewer, doing such things means a straight away rejection of the paper. I think none of the researchers try to do such type of practices. Your statement clearly shows you are biased otherwise any researcher having brain will never do such things that you have blamed us.
  1. The methodology section is still very poor.
  • I have constructed the pilot plant and I am designing such plants from 2008, how it is possible that I have not mentioned all the operational details. If it is poor, you should point the line numbers that are poor. In every paper for review why line number are inserted? It is because a reviewer should tell properly where the mistake is.

In short, there is a lack of convincing evidence to support their novelty in the manuscript. It is obvious the quality of the manuscript does not meet the standards of the water Journal therefore, should be rejected.

  • Worthy reviewer, I am constantly reviewing for reputed journals and from my experience I have learned that if a reviewer is biased, he will always revolve around novelty whereas an expert scientist will technically prove your mistake. Novelty do not have any physical existence that could be proved that’s why a biased reviewer use it as a weapon of revenge.

Round 3

Reviewer 2 Report

The study comprised of “Novel magnetite nanocomposites (Fe3O4/C) for efficient immo-2 bilization of Ciprofloxacin from aqueous solutions through ad-3 sorption and membrane cross-linking processes”. The study was designed to fabricate iron-based magnetic carbon nanocomposites (Fe3O4/C) from mango biomass precursors and utilized as an adsorbent for the ciprofloxacin removal from wastewater through batch adsorption and adsorption/membrane cross-linking processes.

  1. The results shown in this study are lack of innovation and novelty. One of the major flaws is that the experimental design and results are rather simple, making the manuscript lack of strong elucidation to support their novelty.
  2. The literature review is still poor and not strong enough to provide research gaps to prove the novelty of this work. The background of the study is missing, no comparison has been made with the previous data and other renewable energy sources. The identification of the research gaps is still missing.
  3. Furthermore, the results and discussion section still lacks with proper discussion in comparison to the previous studies.
  4. The authors have failed in explaining the results with proper discussion and did not clearly discuss the reasons for trends reported in the manuscript.
  5. Some of the figures and tables are just inserted without proper explanation which is insufficient to present the complete pictures of the present research study.
  6. There is a repetition of sentences having the same concept present throughout the manuscript.
  7. The English language used in the manuscript still needs major improvements as there are many punctuation and grammatical mistakes present throughout the manuscript. Sentences need more clarity and better construction.
  8. The authors have ignored and deleted some of the reviewer’s comments without bothering to address and make the required changes in the revision.
  9. The methodology section is still very poor.

In short, there is a lack of convincing evidence to support their novelty in the manuscript. It is obvious the quality of the manuscript does not meet the standards of the water Journal therefore, should be rejected.

Author Response

Reviewer 2

The study comprised of “Novel magnetite nanocomposites (Fe3O4/C) for efficient immo-2 bilization of Ciprofloxacin from aqueous solutions through ad-3 sorption and membrane cross-linking processes”. The study was designed to fabricate iron-based magnetic carbon nanocomposites (Fe3O4/C) from mango biomass precursors and utilized as an adsorbent for the ciprofloxacin removal from wastewater through batch adsorption and adsorption/membrane cross-linking processes.

  1. The results shown in this study are lack of innovation and novelty. One of the major flaws is that the experimental design and results are rather simple, making the manuscript lack of strong elucidation to support their novelty.
  • The novelty statement was accordingly provided in the last paragraph. Initially we have optimized the adsorbent, then optimized the naked membranes and then adsorption/membrane experiments have been performed. The experiments are in this sequence and results are also presented in this sequence. In my opinion the design is not so simple
  1. The literature review is still poor and not strong enough to provide research gaps to prove the novelty of this work. The background of the study is missing, no comparison has been made with the previous data and other renewable energy sources. The identification of the research gaps is still missing.
  • The available literature has already been cited as far the renewable energy sources are concern, our study deals with foul controlling not with such sources.
  1. Furthermore, the results and discussion section still lacks with proper discussion in comparison to the previous studies.
  • The discussion in last review round was improved accordingly
  1. The authors have failed in explaining the results with proper discussion and did not clearly discuss the reasons for trends reported in the manuscript.
  • The discussion in last review round was improved accordingly
  1. Some of the figures and tables are just inserted without proper explanation which is insufficient to present the complete pictures of the present research study.
  • The required detail was inserted accordingly whereas table 5 was mis placed. Now it has been brought to experimental section accordingly.
  1. There is a repetition of sentences having the same concept present throughout the manuscript.
  • In last review round such redundancy have been removed accordingly.
  1. The English language used in the manuscript still needs major improvements as there are many punctuation and grammatical mistakes present throughout the manuscript. Sentences need more clarity and better construction.

They were corrected accordingly

  1. The authors have ignored and deleted some of the reviewer’s comments without bothering to address and make the required changes in the revision.
  • May be as human error this would have occurred which were accordingly considered in this version.
  1. The methodology section is still very poor.
  • The section was revised accordingly.

This manuscript is a resubmission of an earlier submission. The following is a list of the peer review reports and author responses from that submission.